# Assessment on patient outcomes of primary hip replacement: an interrupted time series analysis from 'The National Joint Registry of England and Wales'

Cesar Garriga [ID],[1,2] Jacqueline Murphy [ID],[1,3] Jose Leal,[3] Nigel K Arden,[1] Andrew James Price [ID],[1] Daniel Prieto-Alhambra,[1,2] Andrew J Carr,[1] Amar Rangan [ID],[1,4] Cyrus Cooper,[1,5] George Peat,[6] Ray Fitzpatrick,[3] Karen L Barker,[1,7] Andrew Judge[1,8]

**Correspondence to**
Dr Cesar Garriga;
cesar.garriga-fuentes@ndorms.ox.ac.uk

## ABSTRACT

**Objectives** Effects of the UK Department of Health's national Enhanced Recovery After Surgery (ERAS) Programme on outcomes after primary hip replacement.

**Design** Natural experimental study using interrupted time series to assess the changes in trends before, during and after ERAS implementation (April 2009 to March 2011).

**Setting** Surgeries in the UK National Joint Registry were linked with Hospital Episode Statistics containing inpatient episodes from National Health Service trusts in England and patient reported outcome measures.

**Participants** Patients aged ≥18 years from 2008 to 2016.

**Main outcome measures** Regression coefficients of monthly means of length of hospital stay, bed day cost, change in Oxford Hip Scores (OHS) 6 months post-surgery, complications 6 months post-surgery and revision rates 5 years post-surgery.

**Results** 438 921 primary hip replacements were identified. Hospital stays shortened from 5.6 days in April 2008 to 3.6 in December 2016. There were also improvements in bed day costs (£7573 in April 2008 to £5239 in December 2016), positive change in self-reported OHS from baseline to 6 months post-surgery (17.7 points in April 2008 to 22.9 points in December 2016), complication rates (4.1% in April 2008 to 1.7% March 2016) and 5 year revision rates (5.9 per 1000 implant-years (95% CI 4.8 to 7.2) in April 2008 to 2.9 (95% CI 2.2 to 3.9) in December 2011). The positive trends in all outcomes started before ERAS was implemented and continued during and after the programme.

**Conclusions** Patient outcomes after hip replacement have improved over the last decade. A national ERAS programme maintained this improvement but did not alter the existing rate of change.

## INTRODUCTION

Osteoarthritis represents a significant, and growing, population health burden worldwide.[1] In the UK, over 1 million adults aged over 45 years consult general practice for osteoarthritis each year.[2] Osteoarthritis accounts for £3.2 billion in lost productivity

## Strengths and limitations of this study

► Our study design is a 'natural experiment' which controlled for unobservable sources of confounding. This approach for evaluating complex interventions allows for causal inferences without randomised controlled experiments.

► Routinely collected data provided actual-practice information on trends in length of stay, patient reported outcome measures of hip pain and function, complications and revision surgeries following primary hip replacement procedures.

► A limitation is the variation in interpretation and adoption across centres because what constitutes Enhanced Recovery After Surgery (ERAS) was not clearly established after the expected identification of best practices in the first year of the ERAS programme.

in the UK,[3] with total direct and indirect costs equivalent to 0.25% to 0.50% of the gross domestic product.[4 5] Almost 88 000 primary hip replacement operations were undertaken during 2016 in the National Health Service (NHS), over 90% of which were for osteoarthritis. This number continues to increase.[6]

Between April 2009 and March 2011, the UK Department of Health implemented an Enhanced Recovery After Surgery (ERAS) Partnership Programme[7] to improve recovery in major planned colorectal, musculoskeletal, gynaecological and urological surgical pathways. The first year of the programme focused on learning best practice from pioneer units of ERAS practice in the NHS. ERAS has a series of evidence-based care elements that all support recovery by reducing the bodily stress reactions caused by injury during surgery. These reductions in the stress responses are of particular importance for

the vulnerable patient with comorbidities, who is often also frail and elderly.[8] It collected information about principles of enhanced recovery, clinical elements of the patient pathway, metrics and success factors. It established a website to share information and resources, generated a financial and equality impact evaluation, published an implementation guide and developed an online reporting tool to support implementation. A lead for enhanced recovery was named in each local health authority to prepare for the spread of the ERAS programme across the NHS during the implementation phase in the programme's second year.

The ERAS in musculoskeletal care focused on planned hip and knee replacement surgeries. ERAS is a complex intervention[9 10] to improve several areas of care across a patient's pathway through surgery. Preoperatively, ERAS aims for the patient to be in the best possible condition for surgery. In primary care haemoglobin levels and pre-existing comorbidities like diabetes are assessed. At the hospital nurses test cardiopulmonary exercise and appropriate anaesthetic for covering surgery is evaluated. In addition, there is informed decision-making after offering to the patient information and managing her/his expectations. At the admission, hospitals arrange to admit patients the same day of surgery, fluid hydration is optimised using oral complex carbohydrates to reduce patient anxiety, reduce the body's resistance to insulin and inflammatory response. Perioperatively, ERAS aims to give the patient the best possible management during and after surgery.[11] It includes: minimally invasive surgery if possible, individualised fluid therapy, avoid crystalloid overload, use of regional/spinal and local anaesthetic with sedation and hypothermia prevention. Postoperatively, ERAS aims to give the patient the best possible rehabilitation. It includes: no routine use of wound drains and/or nasogastric tubes, active, planned mobilisation within 24 hours, early oral hydration and nutrition, intravenous therapy stopped early, catheters removed early, oral analgesia avoiding systemic opiates where possible. Follow-up covers: discharge on planned day or when criteria met, therapy support (stoma, physiotherapy, dietitian…) and 24 hours telephone follow-up if appropriate. Patients are given information before and after surgery, such as changes to make around the home, strengthening exercises and changes to nutrition. ERAS aims to enable earlier return home from hospital with tailored discharge, when suitable for the patient.

Despite their widespread implementation, there is limited evidence about the effectiveness of ERAS programmes in hip and knee replacement surgery.[12] Length of hospital stay has been declining prior to the intervention, and we hypothesised that after the implementation of ERAS, this downward secular trend would decline faster. For the outcomes of complications, revision, pain and function, we did not have a specific a-prior hypothesis as it is unclear what impact ERAS would have on these outcomes. We aimed to determine whether implementing ERAS in hip replacement has led to improved patient outcomes and shorter hospital stays.

## METHODS

### Data source

We used data from the UK National Joint Registry (NJR), which contains data on hip replacement surgeries from all English and Welsh hospitals. It includes 2 million patients since 2003 and covers 95% and 91% of primary hip replacements and revisions, respectively.[13]

### Data linkages

Primary operations were linked with Hospital Episode Statistics (HES) data, which contain records of all NHS funded inpatient episodes undertaken in NHS trusts in England (125 million each year). Planned hip replacements were linked to patient reported outcome measures (PROMs). Patients funded by the NHS in England are asked to complete questionnaires before and 6 months after surgery to evaluate their perceived improvement in health. We retrieved a cohort of patients undergoing planned hip replacement in England between April 2008 and December 2016. Mortality data were matched to the Office for National Statistics database.

### Outcome measures

We evaluated trends in hospital stay length for patients undergoing primary hip replacement. Length of stay was calculated as the number of days between the hospital admission and discharge dates. We used the same set of patients to estimate the inpatient cost for the index episode using NHS reference costs from 2015/2016.[14] We estimated the mean cost per bed day based on each patient's healthcare resource use and length of hospital stay. online supplementary appendix 1 further explains the cost methods.

We also assessed the absolute change in the Oxford Hip Score (OHS), a PROM. Patients complete the same questionnaire about their hip pain and function before and 6 months after surgery.[15] Each question is scored between 0 (worst symptoms) and 4 (least symptoms). The scores from the 12 questions are summed to give a total score between 0 (worst) and 48 (best). We calculated the difference between the total scores 6 months post-surgery and at baseline to obtain a measure of change associated with surgery. Higher positive values for OHS change represent greater self-reported improvement in pain and function.

We calculated the 6 months post-surgery complication proportions. We defined postoperative complications as one or more events happening up to 6 months after primary hip replacement: stroke (excluding transient ischaemic attack), respiratory infection, acute myocardial infarction, pulmonary embolism/deep vein thrombosis, urinary tract infection, wound disruption, surgical site infection, fracture after implant, complication of prosthesis, neurovascular injury, acute renal failure or blood transfusion. We identified these complications in

HES data using diagnosis codes from the 'International Statistical Classification of Diseases and Related Health Problems 10th Revision' (ICD-10) (online supplementary appendix 2), except for blood transfusion, for which we used the 'Classification of Interventions and Procedures version 4' codes (online supplementary appendix 3).

We evaluated the rate of revision surgery up to 5 years after primary hip replacement. We included revisions declared to the NJR registry by surgeons[16] and reported to HES using codes from online supplementary appendix 4. We specified our analysis time in years, reporting the rate as the number of revisions per 1000 implant-years.

### Intervention

The ERAS programme for hip replacement surgery was implemented nationally between April 2009 and March 2011. During the first year the programme focused on identifying best practice, determining clinical elements of the patient pathway, publishing an implementation guide, supporting early adopters of the programme to better understand key factors for implementation and sustainability.[17] During the second year ERAS supported local health areas for delivering and commissioning implementation of ERAS.

### Potential modifiers

We evaluated whether trends in hospital stay length and OHS change after surgery differed by age at the time of primary hip replacement (18 to 59, 60 to 69, 70 to 79, 80 to

84 or ≥85 years) and presence of comorbidities according to the Charlson classification[18] (no comorbidities vs one or more comorbidities): myocardial infarction, congestive heart failure, peripheral vascular disease, cerebrovascular disease, dementia, chronic obstructive pulmonary disease, connective tissue disease, peptic ulcer disease, mild liver disease, mild diabetes, hemiplegia, moderate/severe renal disease, severe diabetes (ie, with organ damage), tumour, leukaemia, lymphoma, moderate/severe liver disease, AIDS and metastatic solid tumour. The ICD-10 codes used are listed in online supplementary appendix 5.

### Participants and inclusion criteria

We included patients receiving planned hip replacement surgery (figure 1) between 1 April 2008 and 31 December 2016. We excluded patients without a concordant date of replacement between NJR and HES registries.

When analysing length of stay, we also excluded patients staying more than 15 days in hospital and patients with a missing hospital stay length or a hospital discharge date before their hospital admission date. When analysing change in OHS, we also excluded patients missing baseline and/or 6 month follow-up OHS scores. When analysing complications, we also excluded patients with complications before the surgery date and those with surgery after June 2016 to guarantee all patients had at least 6 months of follow-up. When analysing 5 year revision rates,

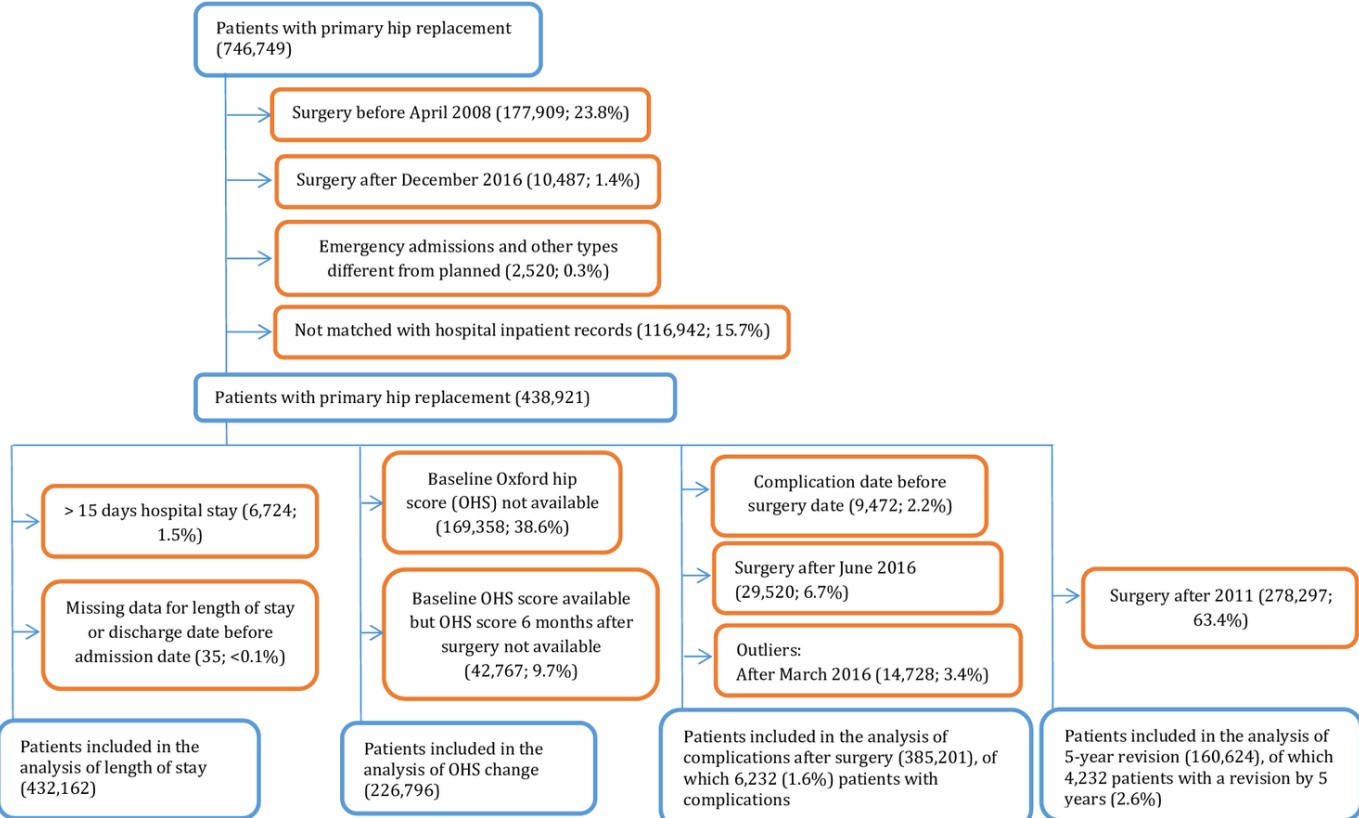

**Figure 1** Flow diagram showing selection of patients for inclusion in this study (blue shows inclusion and orange shows exclusion).

we excluded patients who received surgery after 2011 to ensure all patients had at least 5 years of follow-up.

## Missing data

We excluded patients with missing length of hospital stay or missing OHS data from the respective analyses. We used Pearson's $\chi^2$ statistic to compare missingness in the OHS measure before, during and after ERAS implementation (April 2008 to March 2009, April 2009 to March 2011 and April 2011 to December 2016), across patient age (18 to 59, 60 to 69, 70 to 79, 80 to 84 and ≥85 years) and in the presence and absence of comorbidities. We also compared the distribution of patients with and without OHS data by study period, patient age and presence of comorbidities.

## Study design and statistical analysis

We used a natural experimental study design.[19] We evaluated the effect of ERAS on trends before, during and after its implementation.[20 21] Trusts could choose when to start implementing the programme within the implementation period (April 2009 to March 2011), and we assumed that they only implemented the programme within this period. We described the trends by calculating monthly outcomes as means (length of stay, bed costs, change in OHS), proportions (complications) or rates (revision), and their 95% CIs. We estimated a fractional polynomial over the study period and plotted the resulting curve with the CI of the mean.

We used an interrupted time series approach to estimate changes in outcomes during and immediately after the intervention period, while controlling for baseline levels and trends. We modelled aggregated data points of each outcome of interest by month using segmented linear regression[21]:

$$Y_t = \beta_0 + (\beta_1 * \text{time}_t) + (\beta_2 * \text{ERAS}_0) + (\beta_3 * \text{time after ERAS}_0) + (\beta_4 * \text{ERAS}_{end}) + (\beta_5 * \text{time after ERAS}_{end}) + e_t.$$

$Y_t$ is the mean of each outcome for patients undergoing primary hip replacements – mean days in hospital for length of stay, mean OHS change for the PROM analysis, mean proportion of complications for 6 month complications outcome and mean rate of revisions of primary hip replacements for the 5 year revision outcome – in month t. 'Time' is a continuous variable representing the number of months from the start of the observation period (April 2008) until time t. $\beta_0$ estimates the baseline level of the outcome at the beginning of the time series (April 2008). $\beta_1$ estimates the trend before ERAS was implemented in April 2009. $\beta_2$ is the change in level immediately after the intervention (ERAS$_0$ = April 2009). $\beta_3$ estimates the change in the trend in the monthly mean (number or rate, depending on outcome) after ERAS started (ie, ERAS implementation trend). $\beta_4$ is the change in level immediately after the end of the intervention (ERAS$_{end}$ = March 2011). $\beta_5$ estimates the change in the trend in the mean monthly number or rate (depending on outcome) after ERAS ended.

We excluded non-significant terms using a backward approach to maximise statistical power, producing a parsimonious model with meaningful selected variables. In preliminary analysis we checked for autocorrelation with the previous month, 2 months…12 months using Durbin's alternative test.[22] As autocorrelation invalidated the interpretation of the model, we estimated the linear regression models with Newey-West standard errors.[23] Parsimonious models were generated using the variables previously selected in the backward regression. We also report the full models.

All analyses were conducted using Stata V.13.1 statistical software (StataCorp, College Station, Texas). We followed the STROBE (Strengthening the Reporting of Observational Studies in Epidemiology) guideline in reporting this study.[24]

## Patient and public involvement

The James Lind Alliance has identified the need for involving patients in identifying outcomes that matter to them (patient-identified outcomes). The study has been developed in collaboration with the University of Bristol, Musculoskeletal Research Unit's Patient Experience Partnership in Research (PEP-R) group to identify outcomes from those available in the routine data sets available for this study. The top outcomes were: (1) pain and function, (2) complications (particularly hospital-acquired infection), (3) length of stay (dependent on the level of support at home), (4) revision surgery and (5) mortality (rated low importance) (further detail is provided in online supplementary text S1).

## RESULTS

We identified 438 921 planned primary hip replacements between April 2008 and December 2016 (figure 1). Sixty per cent of patients were women and the average age was 69 years (SD ±11 years). The mean body mass index (BMI) at primary surgery of 28.9 kg/m$^2$ (SD ±5.2 kg/m$^2$) fell into the overweight category,[25] although BMI was missing for 28% of patients. Most patients (84%) had a physical status[26] of 'mild' or 'fit'.

### Length of stay

Hospital stays shortened from 5.6 days (95% CI 5.5 to 5.7) in April 2008 to 3.6 (95% CI 3.6 to 3.7) in December 2016 (figure 2A). As shown in table 1, they were already shortening significantly by −0.020% every month (95% CI −0.023% to −0.017%) before ERAS (full models in online supplementary table 1S1). Hospital stay shortened more quickly (−0.033%) during the implementation period (April 2009 to March 2011), then declined at a slower rate (−0.002%) once ERAS ended (April 2011 to December 2016).

Although older patients had longer hospital stays, all age groups shared the shortening stay trend (figure 3, online supplementary tables S2,S3). For example, those aged 18 to 59 years decreased from 4.7 days (95% CI 4.6 to 4.9) in

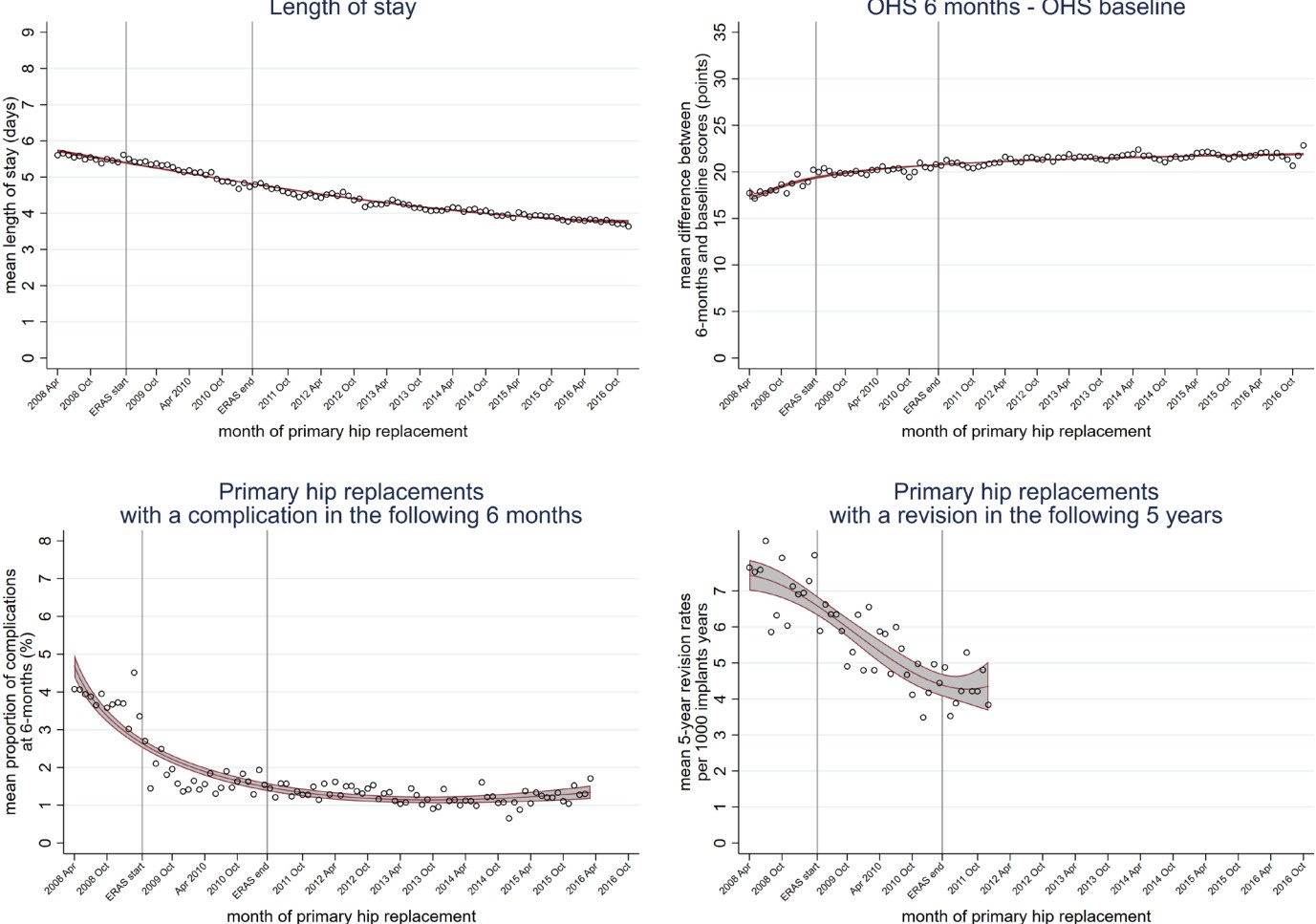

**Figure 2** Effect of the enhanced recovery after surgery (ERAS) programme implemented from April 2009 to March 2011 on trends in outcomes following primary hip replacement in England, UK, 2008 to 2016, by month. (A) length of hospital stay, (B) change in self-reported pain and function, measured using the Oxford Hip Score (OHS) at baseline and 6 months after the surgery, (C) any complication in the 6 months after surgery and (D) hip revision in the 5 years after surgery.

April 2008 to 3.0 days (95% CI 2.9 to 3.1) in December 2016 and those aged ≥85 years decreased from 8.1 days (95% CI 7.7 to 8.6) in April 2008 to 5.3 days (95% CI 4.9 to 5.6) in December 2016. Patients with and without pre-existing comorbidity also had shortening hospital stays (figure 4).

We estimated cost data for 432 143 patients. The mean inpatient bed day cost over time showed a similar trend to that observed for hospital stay length. The overall mean cost of the index hospital episode decreased from £7573 (95% CI £7477 to £7668) in April 2008 to £5239 (95% CI £5171 to £5306) in December 2016 (online supplementary figure S1).

### OHS change
We excluded 48% of patients with missing OHS information from the change in OHS analysis (figure 1). We found more missing OHS data before the intervention (89.7%) than during (41.9%) or after (45.0%) the intervention (online supplementary table S4). online supplementary table S5 shows more patients without OHS change data than with this data in the period prior to ERAS (16.2% and 1.7%, respectively).

Self-reported OHS scores improved over the study period, with an increase in OHS 6 months after surgery of 17.7 points (95% CI 16.4 to 19.0) in April 2008, to 22.9 points (95% CI 21.8 to 23.9) in December 2016 (figure 2B). This trend was also seen in patients with and without comorbidities and in all age groups, except in those aged ≥85 years, whose change in OHS remained stable over the study period (online supplementary figures S2,S3, online supplementary tables S6,S7).

According to the interrupted time series model (table 1), OHS change increased significantly by 0.158% (95% CI 0.130% to 0.186%) every month before the intervention (see online supplementary table S1 for the full model). During ERAS implementation (April 2009 to March 2011), the increase continued but slowed to 0.027%. Change in OHS then became stable after ERAS implementation ended (April 2011 to December 2016).

### Complications at 6 months
Six thousand two hundred and thirty-two (1.6%) patients with a primary hip replacement between April 2008 and March 2016 had one or more complication in the

**Table 1** Temporal trends in patients undergoing planned primary hip replacement from April 2008 to December 2016, parsimonious models with Newey-West standard errors

| Parameter | Coefficient | Lower 95% CI | Upper 95% CI | P value |
|---|---|---|---|---|
| *Length of stay in hospital* | | | | |
| Intercept | 5.674 | 5.655 | 5.693 | <0.001 |
| Monthly trend | −0.020 | −0.023 | −0.017 | <0.001 |
| Level change $ERAS_0$ | 0.176 | 0.120 | 0.232 | <0.001 |
| Trend change after $ERAS_0$ | −0.013 | −0.017 | −0.009 | <0.001 |
| Level change $ERAS_{end}$ | −0.102 | −0.203 | −0.001 | 0.049 |
| Trend change after $ERAS_{end}$ | 0.019 | 0.015 | 0.022 | <0.001 |
| *Change in Oxford hip score (score at 6 months – score at baseline)* | | | | |
| Intercept | 17.063 | 16.896 | 17.230 | <0.001 |
| Monthly trend | 0.158 | 0.130 | 0.186 | <0.001 |
| Level change $ERAS_0$ | 0.772 | 0.538 | 1.006 | <0.001 |
| Trend change after $ERAS_0$ | −0.131 | −0.161 | −0.101 | <0.001 |
| Level change $ERAS_{end}$ | 0.564 | 0.208 | 0.920 | 0.002 |
| Trend change after $ERAS_{end}$ | −0.013 | −0.025 | −0.001 | 0.039 |
| *Complication by 6 months after surgery* | | | | |
| Intercept | 4.044 | 3.465 | 4.624 | <0.001 |
| Monthly trend | −0.078 | −0.096 | −0.061 | <0.001 |
| Level change $ERAS_0$ | — | — | — | — |
| Trend change after $ERAS_0$ | — | — | — | — |
| Level change $ERAS_{end}$ | — | — | — | — |
| Trend change after $ERAS_{end}$ | 0.078 | 0.056 | 0.100 | <0.001 |
| *Revision rates by 5 years after surgery* | | | | |
| Intercept | 7.901 | 7.653 | 8.149 | <0.001 |
| Monthly trend | −0.098 | −0.108 | −0.087 | <0.001 |
| Level change $ERAS_0$ | — | — | — | — |
| Trend change after $ERAS_0$ | — | — | — | — |
| Level change $ERAS_{end}$ | — | — | — | — |
| Trend change after $ERAS_{end}$ | 0.091 | 0.052 | 0.129 | <0.001 |

Confidence intervals, CI; Enhanced Recovery After Surgery, ERAS; start point of ERAS intervention in April 2009, $ERAS_0$; end point of ERAS intervention in March 2011, $ERAS_{end}$; —, p≥0.05.

6 months after surgery. The proportion of complications at 6 months decreased from 4.1% (95% CI 3.4 to 4.7) to 1.7% (95% CI 1.3 to 2.1) over the study period (figure 2C). According to the interrupted time series model, complications at 6 months decreased by −0.078% (95% CI −0.096% to −0.071%) every month before the intervention (table 1). During the intervention, the trend reversed and complications increased by 0.078%. The proportion of complications stabilised after the intervention ended.

### Five year revision rates

According to the NJR registry, 3392 (2.1%) patients with a primary hip replacement between April 2008 and December 2011 had a hip revision in the following 5 years. We found 840 more 5 year revisions using HES, giving a total of 4232 (2.6%). Rates of 5 year hip revision decreased from 7.6 per 1000 implant-years (95% CI 6.4 to 9.2) at risk in April 2008 to 3.8 (95% CI 3.0 to 4.9) in December 2011 (figure 2D).

According to a parsimonious model of 5 year hip revision rates (table 1), there was a significant downward trend of −0.098 per 1000 implant-years (95% CI −0.108 to −0.087) before the implementation of the intervention (April 2009 to March 2011) (see online supplementary table S1 for the full model). The trend reversed after the intervention ended (April 2011 to December 2016) to increase 0.091 per 1000 implant-years (95% CI 0.052 to 0.129).

# Length of stay

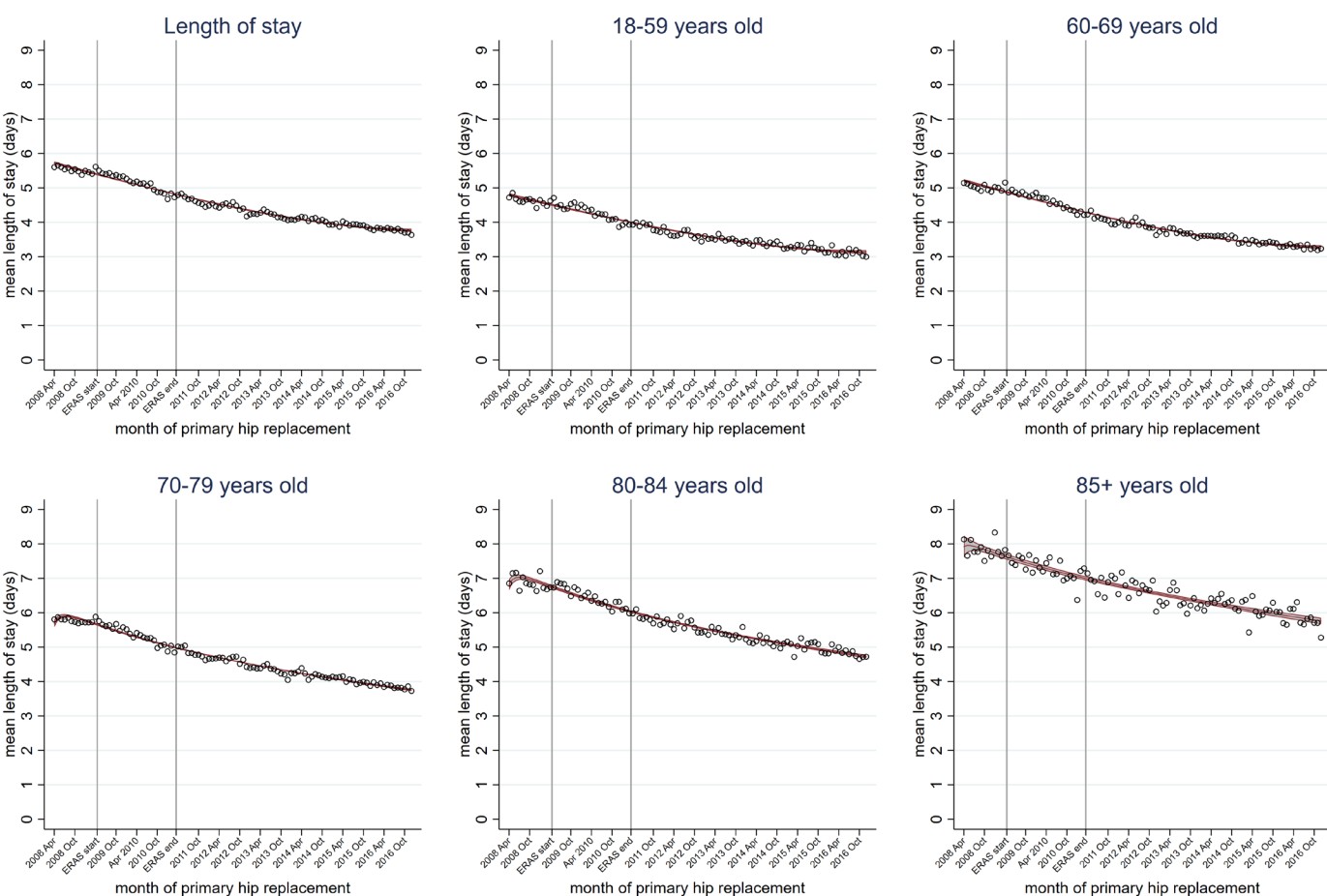

**Figure 3** Trends in length of hospital stay following primary hip replacement according to patient age categories in England, UK, 2008 to 2016, by month. Enhanced recovery after surgery (ERAS) programme implemented in England from April 2009 to March 2011.

## DISCUSSION

We have shown that primary hip replacement outcomes have been improving since 2008, with decreasing length of hospital stay, estimated average inpatient bed day costs, complications and 5 year revision risk, and improving patient-reported pain and function. These positive trends were seen across all age groups and in those with and without comorbidity, and began before the NHS ERAS programme was implemented. Hospital stays have shortened without adversely affecting patient outcomes.

We hypothesised that the national ERAS intervention would improve primary hip replacement outcomes by changing trends during and after its implementation. Our hypothesis was not confirmed, as ERAS did not influence existing trends. However, we collected only 1 year of data before the intervention was implemented (April 2008 to March 2009), in comparison with 2 years of data during implementation (April 2009 to March 2011) and 5 years of data post-intervention (April 2011 to December 2016). We know from other UK studies that length of stay has been in gradual decline in the years prior to 2008, where Burn *et al* found that in 1997 mean hospital stays

for total hip replacement was 14.28 days, and in 2008, before the ERAS intervention, 7.94 days[27].

Although our study design controlled for unobservable sources of confounding, we stratified by age and presence/absence of comorbidities at surgery to detect patterns in outcome variation. This approach for evaluating complex interventions allows for strong causal inferences without randomised controlled experiments.[10 20]

For this 'natural experiment', we assumed that ERAS was implemented homogenously across all England NHS trusts in the 2 year implementation period. This assumption is unlikely to be true. The shortening hospital stays and improved outcomes seen before April 2009 may reflect some trusts implementing ERAS elements before the national programme began. Shortening hospital stays before ERAS may also reflect attempts to improve the cost-effectiveness of hip replacement surgery, which is an important expenditure for the NHS.[27–29]

The hospitals implemented ERAS on different dates, and some had not implemented ERAS by March 2011.[17] The Department of Health surveyed trusts on their use of ERAS near the end of the implementation period

## Length of stay

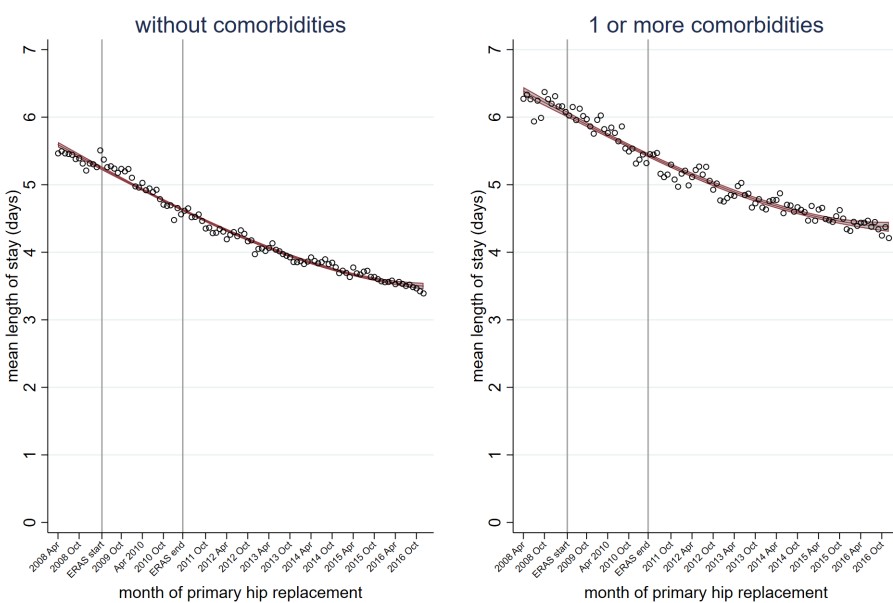

**Figure 4** Trends in length of hospital stay following primary hip replacement according to whether patients do or do not present with comorbidities, in England, UK, 2008 to 2016, by month. Enhanced recovery after surgery (ERAS) programme implemented in England from April 2009 to March 2011.

(February 2011) and reported full implementation by 81 consultant teams, partial implementation by about 20 teams and plans to implement ERAS by about 30 teams. As the Department of Health did not clearly describe ERAS after best practices were identified in the first year of the programme,[30] there is also variation in the teams' interpretation and adoption of the programme. The Department of Health guideline did not indicate how long ERAS should be implemented for and its later report did not measure actual implementation periods in each hospital.[17 30] Considering the complexity of the intervention and stakeholders involved, implementation length is likely to have varied between hospitals. Our quasi-experimental approach smooths the potential dissimilarities.

Existing systematic reviews and randomised clinical trials have found that ERAS programmes for planned colorectal, joint replacement and other surgeries reduced hospital stays when compared with conventional care.[12] Many kinds of ERAS programmes for hip replacement have been investigated: physical therapy the same day of the surgery in the recovery room[31]; preoperative patient education, postoperative multimodal analgesia with periarticular injections, early physiotherapy and rehabilitation and discharge home with an outpatient rehabilitation programme[32 33]; patient and staff education on 'enhanced recovery' principles, preadmission medication, perioperative urinary catheterisation, low-dose spinal anaesthesia and aiming for same-day mobilisation[34]; perioperative information, pain relief, nausea control, nutrition, mobilisation and elimination[35]; preoperative

patient seminar, treatment of pain (spinal anaesthesia) and early mobilisation, standardised programme in the operating theatre (tranexamic acid and no drains), 1 to 2 hours of a multimodal fast-track rehabilitation regime, daily physiotherapy within the first 24 hours and a multimodal oral opioid-sparing analgesia[36]; or a perioperative analgesic blocking the peripheral nerve.[37] However, these studies involved small sample sizes (170, 57, 1256, 630, 28, 98, 15),[31–37] were limited to one hospital or trust and only compared their intervention with traditional management. They cannot be generalised to the wider population. We investigated whether the ERAS for hip replacement was successfully implemented by comparing with a pre-ERAS period, as was done in other studies,[32–34] but for the first time, by also comparing with the post-intervention period. We also included all of the hospitals in one country.

Shorter hospital stays were reflected in lower estimated average inpatient bed day costs. Most surgery episodes in the data set had a hospital stay shorter than the trim point for the cost of the relevant healthcare resource use group. A drop in hospital stay length within the trim point would not be reflected by a change in the estimated average episode costs, if we assign the same unit cost to all patients in the same healthcare resource use group who had a length of stay shorter than the trim point. We therefore estimated the true reduction in NHS expenditure by estimating a cost per bed day reflecting each patient's hospital stay length.

OHS change scores increased across the study period: the difference between pain and function at baseline

and 6 months post-surgery was greater at the end of 2016 than in 2008, indicating less pain and better function after surgery. A review of ERAS in total hip replacement showed that better improvement in pain and function scores could be related to making patients active participants in their recovery and managing patient expectations.[29] A Cochrane review on preoperative education for hip or knee replacement do not find benefits over usual care, but did find a non-significant reduction in pain and better function associated with perioperative education.[38]

Complications 6 months after surgery were decreasing before ERAS was implemented, remained steady during the ERAS period and increased after the intervention ended. Discharging patients too soon after surgery could increase complications. A meta-analysis of ERAS programmes for colorectal surgery did not find evidence of increased surgical site infections or anastomotic leakage, classified as surgical complications (relative risk=0.76, 95% CI 0.54 to 1.08), and found that cardiovascular, pulmonary and infectious medical complications decreased (relative risk=0.40, 95% CI 0.27 to 0.61).[39] Patients with diabetes undergoing hip and knee replacement under ERAS protocols had a lower additional risk for complications otherwise associated with operating on patients with diabetes.[40]

The 5 year revision surgery rates dropped over the study period, a desirable finding as the revision procedure is more complicated than the initial procedure.[41] Revision rates may have declined due to the UK National Institute for Health and Care Excellence recommendation to only use implants with a 10 year revision rate of 5% or lower, to avoid low-quality prostheses.[28]

The promising improvements in primary hip replacement outcomes reported here were achieved amidst increasingly strained NHS funding and hospital budgets. NHS funding growth in our study period was slower than historical trends.[42] There are fewer hospital beds available for hip replacement today than at the beginning of the study period, and wards have been closed. For example, there was an average daily 10 015 occupied beds open overnight for trauma and orthopaedics in England between April 2010 and June 2010, dropping to 8770 between October 2016 and December 2016.[43] Conversely, the number of primary hip replacements in England increased from 67 128 in 2008 to 87 733 in 2016.[6] It has been estimated that 97 516 total hip replacements will take place in 2035.[44] Meeting this demand with existing or lower capacity will require efficiencies. During the period of our study we observed an increasing trend in the proportion of NHS funded primary hip replacements being carried out in independent hospitals (increasing from around 10% in 2008 to 27% in 2016) and a small increase in those within Independent Sector Treatment Centres (from 3.5% in 2008 to 5% in 2016). These changes will have supported an increase in capacity for surgery (although such centres typically treat healthier and less complex patients than nearby public hospitals, with a worsening case-mix of those patients treated in public hospitals).[45] Such changes in the sorting of routine and complex patients between public and private hospital settings over time could also influence observed changes in outcomes of surgery over time. ERAS has kept the improvements happening when other changes were occurring which may have caused deterioration, for example, older, sicker and more obese patients. However, changes in the case-mix of patients have not altered improving trends in outcomes of surgery.

An important issue is the high variation in services and practices across English hospitals. The Getting It Right First Time (GIRFT) programme aims to reduce discrepancies in activity volume, implant choice and guideline follow-up between hospitals.[46] Despite national improvements in hip replacement outcomes, GIRFT reports substantial variation in outcomes between hospital trusts. In 2016, the mean length of hospital stay after primary hip replacement varied between 2.5 and 11.6 days, and OHS change varied between 12.0 and 23.5 points. Although the national picture has improved for patients, there is still work remaining to understand and reduce unwarranted outcome variations between individual hospitals. This study would be strengthened with the comparison with trends from another country (Wales for example). However, we do not have access to hospital admission data for Wales for this study. In addition, the comparison with another procedure where the enhanced recovery intervention was not applied, could also act as a potential control group (such as cataract surgery). Nevertheless, the importance of an external control group would have been higher if within this study we had observed a significant impact of ERAS intervention on a change in trend in outcomes of surgery, as it would have given reassurance of an intervention effect was not observed in the control group. However, in the case of our study, we do not observe an intervention effect — rather than continuation of an existing secular trend that was happening before introduction of the intervention.

Although the NJR registry captures all primary hip replacements including those undertaken in the private sector linkage to English HES data means that we only have access to information on patients receiving NHS funded operations including public and private hospitals. Therefore, this study do not include private funded operations undertaken by the independent sector. It is estimated 13.7% to 19.7% of all hip replacements were carried by the independent sector in 2012 to 2013 and 2016 to 2017, respectively (Source: Hospital Episode Statistics, NHS Digital.)

## CONCLUSION

Our study shows that outcomes after planned hip replacement are better today than 10 years ago. Hospital stays shortened substantially from 2008 to 2016 for all age groups and in people with and without comorbidity, without adversely affecting patient outcomes. Patient-reported pain and function have improved, revision rates

are in decline and complication rates remain stable. The introduction of a national ERAS programme maintained this improvement but did not alter the rate of change already underway.

**Author affiliations**

$^1$Nuffield Department of Orthopaedics, Rheumatology and Musculoskeletal Sciences (NDORMS), University of Oxford, Oxford, UK

$^2$Centre for Statistics in Medicine, Nuffield Department of Orthopaedics, Rheumatology and Musculoskeletal Sciences, University of Oxford, Headington, UK

$^3$Nuffield Department of Population Health, University of Oxford, Oxford, UK

$^4$Department of Health Sciences, University of York, York, UK

$^5$MRC Lifecourse Epidemiology Unit, University of Southampton, Southampton, UK

$^6$Primary Care and Health Sciences, Keele University, Keele, UK

$^7$Oxford University Hospitals NHS Foundation Trust Nuffield Orthopaedic Centre, Oxford, UK

$^8$Musculoskeletal Research Unit, Translational Health Sciences, Bristol Medical School, University of Bristol, Bristol, UK

**Acknowledgements** We would like to thank the patients and staff of all the hospitals in England and Wales who have contributed data to the National Joint Registry (NJR); and the Healthcare Quality Improvement Partnership (HQIP), the NJR Steering Committee and staff at the NJR Centre for facilitating this work. The authors have conformed to the NJR's standard protocol for data access and publication. We acknowledge English language editing by Dr Jennifer A de Beyer (Centre for Statistics in Medicine, University of Oxford) and the addition of her informative suggestions. We acknowledge Albert Prats Uribe and Sam Hawley of the Botnar Research Centre (University of Oxford) for providing their expertise in how to conduct an interrupted-time series analysis.

**Contributors** AJ conceived the study. AJ, JL and CG contributed to study design. AJ acquired raw data from NJR, HES and PROMs registries. CG had full access to all of the study data and takes responsibility for the integrity of the data and the accuracy of the data analysis. JM and JL had full access to all of the study data and takes responsibility for the integrity of the generated health-economic dataset and the accuracy of the health-economic data analysis. CG, JM, JL, NA, AP, DPA, AC, AR, CC, GP, RF, KB and AJ contributed to interpretation of results. CG wrote the first draft of the manuscript and AJ made substantial contributions to the final manuscript. CG produced all tables, figures, supplementary material and appendixes except for Appendix 1, JM wrote Appendix 1. CG, JM, JL, NA, AP, DPA, AC, AR, CC, GP, RF, KB and AJ had full access to all statistical reports and tables in the study. CG, JM, JL, NA, AP, DPA, AC, AR, CC, GP, RF, KB and AJ contributed to the interpretation of results and critical revision of the manuscript and approved the final manuscript. AJ is the guarantor. The corresponding author (CG) attests that all listed authors meet authorship criteria and that no others meeting the criteria have been omitted.

**Funding** This project was funded by the NIHR Health Services and Delivery Research programme (project number 14/46/02). Andrew Judge is supported by the NIHR Biomedical Research Centre at the University Hospitals Bristol NHS Foundation Trust and the University of Bristol. The views expressed in this publication are those of the authors and do not necessarily reflect those of the NHS, the National Institute for Health Research or the Department of Health and Social Care.

**Competing interests** All authors have completed the Unified Competing Interest form at www.icmje.org/coi_disclosure.pdf and declare: AJ has received consultancy fees from Freshfields Bruckhaus Deringer, and has held advisory board positions (which involved receipt of fees) from Anthera Pharmaceuticals, INC. AP reports personal fees from Zimmer Biomet, outside the submitted work. AR reports grants from DePuy Ltd outside the submitted work. CC received personal fees from: Alliance for Better Bone Health; Amgen; Eli Lilly; GSK; Medtronic; Merck; Novartis; Pfizer; Roche; Servier; Takeda; UCB, outside the submitted work. NKA received grants and personal fees outside the submitted work from: Bioberica; Merck; Flexion; Regeneron; Freshfields Bruckhaus Deringer. DPA received grants from: Amgen; UCB Biopharma; Servier; Astellas; Novartis, outside the submitted work. All other authors declare no conflicts of interest.

**Patient consent for publication** Not required.

**Provenance and peer review** Not commissioned; externally peer reviewed.

**Data availability statement** Data may be obtained from a third party and are not publicly available. Access to data is available from the National Joint Registry for England and Wales, Northern Ireland and the Isle of Man, but restrictions apply to the availability of these data, which were used under license for the current study, and so are not publicly available. Data access applications can be made to the National Joint Registry Research Committee. Access to linked HES and PROMs data is available through data applications to NHS Digital.

**ORCID iDs**

Cesar Garriga http://orcid.org/0000-0001-7073-3611

Jacqueline Murphy http://orcid.org/0000-0002-3927-7002

Andrew James Price http://orcid.org/0000-0002-4258-5866

Amar Rangan http://orcid.org/0000-0002-5452-8578

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
