## [Reviewer comments · BMJ Open]

ARTICLE DETAILS

TITLE (PROVISIONAL)	Assessment on Patient Outcomes of Primary Hip Replacement: an Interrupted Time Series Analysis from “The National Joint Registry of England and Wales”
AUTHORS	Garriga, Cesar; Murphy, Jacqueline; Leal, Jose; Arden, Nigel; Price, Andrew; Prieto-Alhambra, Daniel; Carr, Andrew; Rangan, Amar; Cooper, Cyrus; Peat, George; Fitzpatrick, Ray; Barker, Karen L.; Judge, Andrew

VERSION 1 – REVIEW

REVIEWER	Bheeshma Ravi University of Toronto, Canada
REVIEW RETURNED	04-Jul-2019

GENERAL COMMENTS	This is a well written and very interesting study. However, I have some concerns/suggestions that might strengthen the manuscript. 1) I think it would be worthwhile to examine a longer period before ERAS implementation - as it stands it is unclear what the 'trend' was before surgery. 2) Please offer a more detailed description of ERAS for those of us that do not work in the NHS. Include reasons for its implementation, as additional justification for the outcomes you have chosen. Could the ERAS have strengthened the changes that were already underway, or was it a non-entity? 3) Can you offer an explanation for why you excluded patients with an LOS >15 days? Why not 5 days? Was this an arbitrary cut-off? 4) Can you expand on additional factors that may have changed over the study period that may have affected outcomes? Changes in surgical approach or patient demographics, etc?
---

REVIEWER	THOMAS WAINWRIGHT Bournemouth University UK
REVIEW RETURNED	25-Jul-2019

GENERAL COMMENTS	Thank you for inviting me to review this work. The authors have done an excellent job and I think that this is an important research question which the authors have answered thoroughly. I have one point that I feel is important to make clear and to be addressed, in both the methods, and in the discussion on outcomes. During the time period studied, there has been an
---

	increasing amount of NHS hip replacements performed in non-NHS settings as either waiting list initiative work, or through patient choice and the AQP route. Whilst these pts may still be registered on the NJR they will not to my knowledge appear in HES statistics. Please can the authors be explicit on how this has been handled within the data, and also consider the effects on outcomes that this may of had in the discussion. If this point is addressed I would recommend for publication.
--	--

VERSION 1 – AUTHOR RESPONSE

Reviewer: 1

Reviewer Name

Bheeshma Ravi

Institution and Country

University of Toronto, Canada

Please state any competing interests or state 'None declared':

None declared

Please leave your comments for the authors below

This is a well written and very interesting study. However, I have some concerns/suggestions that might strengthen the manuscript.

1) I think it would be worthwhile to examine a longer period before ERAS implementation - as it stands it is unclear what the 'trend' was before surgery.

Author response: Our data available to us for this study was only from 2008 onwards, the reason being that data on our PROMs outcome was only available from this date, so all of our HES-PROMs linked data is from 2008. However, we know from other UK studies that length of stay has been in gradual decline in the years prior to 2008, where Burn et al found that in 1997 mean LOS in days for THR was 14.28 and in 2008, before the ERAS intervention, was 7.94 (see Supplementary appendix 1 of that paper).¹

Author action: In the discussion section, we have cited literature regarding patterns in trends of length of stay prior to 2008 (page 16, line 373).

2) Please offer a more detailed description of ERAS for those of us that do not work in the NHS.

Include reasons for its implementation, as additional justification for the outcomes you have chosen.

Could the ERAS have strengthened the changes that were already underway, or was it a non-entity?

Author response: ERAS is a new approach to the preoperative, intraoperative and postoperative care of patients undergoing surgery. Originally pioneered in Denmark² it is now being introduced in England by a growing number of surgeons, anaesthetists, nurses, allied health professionals and NHS managers. The ERAS protocols have a series of evidence-based care elements that all support recovery by reducing the bodily stress reactions caused by injury. These reductions in the stress responses are of particular importance for the vulnerable patient with co-morbidities, who is often also frail and elderly.³

To inform the list of important outcomes for this study, we conducted a forum with the University of Bristol's Musculoskeletal Research Unit's (MRU) patient involvement group: the 'Patient Experience Partnership in Research' (PEP-R). PEP-R comprises twelve patients with musculoskeletal conditions, most of whom have had joint replacement, all of whom have had experience of long-term pain. The group were given a list of outcomes for consideration and discussion that were available in the routine datasets: length of stay, readmission, reoperation, revision surgery, complications, mortality, Oxford hip and knee scores on pain and function. Patients considered pain and function to be the most important outcome, followed by complications of surgery. Length of stay was considered a mid-

ranking outcome, with the group agreeing that it was important but very dependent upon the level of support at home. Revision surgery was also a mid-ranking outcome for the group. Mortality was ranked low by the group in respect of its importance to them, and hence has not been included in the analysis for this paper.

In respect of the reviewers comment regarding the inclusion of reasons for ERAS implementation, as additional justification for the outcomes, enhanced recovery improves quality of care by helping patients to get better sooner after major surgery, that in turn reduces length of stay with benefits to the NHS.⁴ Length of stay is a proxy for efficiency of health care provision in the absence of data on clinical quality.⁵⁻⁷ In addition, length of stay is the outcome used for the UK Department of Health to report results of ERAS implementation.⁸ Complications are a measure of efficacy and safety of ERAS implementation. Revision surgery was the main method for assessing outcomes of hip replacement surgery until patient reported outcome measures were introduced in 2008.

Regarding the reviewers comments as to whether ERAS could have strengthened the changes that were already underway, we would suggest that an important consideration of the findings of this study are that the secular trends of improved outcomes have continued following the introduction of ERAS, rather than stopping. These improvements have occurred whilst other changes have been occurring, including reductions in the numbers of available beds/wards/operating theatres, in addition to increasing absolute numbers of patients undergoing THR year on year. Further the case mix of patients has also been changing over the study time period. In our dataset we find a higher percentage of sicker patients in the intervention period (April 2009-March 2011) and even that is higher again in the post intervention period (April 2011-December 2016) than preintervention period (April 2008-March 2009). In the discussion section we have attempted to comment on these "external influencing factors" (pages 19-20, lines 454-474).

Author action: we have added to the text further explanation of ERAS. Page 5, lines 100-103: "ERAS has a series of evidence-based care elements that all support recovery by reducing the bodily stress reactions caused by injury during surgery. These reductions in the stress responses are of particular importance for the vulnerable patient with co-morbidities, who is often also frail and elderly."

We also provide further detail about ERAS pathway. Page 6, lines 114-120: "In primary care haemoglobin levels and pre-existing co-morbidities like diabetes are assessed. At the hospital nurses test cardiopulmonary exercise and appropriate anaesthetic for covering surgery is evaluated. In addition, there is informed decision making after offering to the patient information and managing her/his expectations. At the admission, hospitals arrange to admit patients the same day of surgery, fluid hydration is optimised using oral complex carbohydrates to reduce patient anxiety, reduce the body's resistance to insulin and inflammatory response." Page 6, lines 121-124: "It includes: minimally invasive surgery if possible, individualised fluid therapy, avoid crystalloid overload, use of regional/spinal and local anaesthetic with sedation, and hypothermia prevention." Page 6, lines 125-129: "It includes: no routine use of wound drains and/or nasogastric tubes, active, planned mobilisation within 24 hours, early oral hydration and nutrition, intravenous therapy stopped early, catheters removed early, oral analgesia avoiding systemic opiates where possible. Follow-up covers: discharge on planned day or when criteria met, therapy support (stoma, physiotherapy, dietitian...) and 24 hour telephone follow-up if appropriate."

3) Can you offer an explanation for why you excluded patients with an LOS >15 days? Why not 5 days? Was this an arbitrary cut-off?

Author response: There were very few patients with a LOS > 15 days (around 1.5% of patients stayed longer than 15 days). We made a decision to exclude them, as we were concerned that leaving those patients in the analysis could create statistical "noise" in the observed trends. Discharge delaying more than 15 days might be for other reasons beyond the hip replacement surgery. 5 days as a cut-off would exclude patients with comorbidities, older age and frailty. E.g. in our results patients aged ≥85 years stayed 8.1 days (95% CI: 7.7 to 8.6) in April 2008.

Author action: None

4) Can you expand on additional factors that may have changed over the study period that may have affected outcomes? Changes in surgical approach or patient demographics, etc?

Author response: Improvements in outcomes of surgery have occurred whilst other changes have been occurring with an increasing absolute numbers of patients undergoing hip replacement year on year. Further the case mix of patients has also been changing over the study time period. In our dataset we find an increased percentage of older patients (>80 years old) undergoing hip replacement (see Response Figure R1 below). We also observe an increasing percentage of sicker patients (see

ascending trends or percentage of patients with mild, moderate and severe Charlson co-morbidity index in Response Figure R2). Similar increasing trends are observed: in obese patients (Response Figure R3); in patients with a pre-operative incapacitating disease or life-threatening disease (Response Figure R4) and patients undergoing a surgical posterior approach (Response Figure R6). Percentages of IMD categories (Response Figure R5) are steady across the study period except for those living in the most deprived areas with a decreasing percentage of surgeries. Response Figure R1. Trends in percentage of age category following primary hip replacement, in England, UK, 2008 – 2016, by month.

% of age category by month in hip replacement

Enhanced recovery after surgery programme implemented in England from April 2009 to March 2011, ERAS.

Response Figure R2. Trends in percentage of Charlson co-morbidity index category following primary hip replacement, in England, UK, 2008 – 2016, by month.

% of co-morbidities index category by month in hip replacement

Enhanced recovery after surgery programme implemented in England from April 2009 to March 2011, ERAS.

Response Figure R3. Trends in percentage of BMI category following primary hip replacement, in England, UK, 2008 – 2016, by month.

% of BMI category by month in hip replacement

Body mass index, BMI; enhanced recovery after surgery programme implemented in England from April 2009 to March 2011, ERAS. Underweight $\leq 18.5 \text{ Kg/m}^2$, Normal $> 18.5 \text{ Kg/m}^2$ to $\leq 25 \text{ Kg/m}^2$, Overweight $> 25 \text{ Kg/m}^2$ to $\leq 30 \text{ Kg/m}^2$, Obese Class I (Moderately obese) $> 30 \text{ Kg/m}^2$ to $\leq 35 \text{ Kg/m}^2$, Obese Class II and higher $> 35 \text{ Kg/m}^2$

Response Figure R4. Trends in percentage of pre-operative ASA physical function score category following primary hip replacement, in England, UK, 2008 – 2016, by month.

A patient’s physical status is classified by the American Society of Anesthesiologists (ASA) grade. This grading system is a standard assessment of the patient’s general physical health prior to surgery (<https://doi.org/10.4103/0019-5049.79879>). It is composed by five categories (1, fit and healthy; 2, mild disease; 3, incapacitating disease; 4, life-threatening disease; and 5 expected to die within 24 hours). Categories 3 to 5 are lumped in one graph. Enhanced recovery after surgery programme implemented in England from April 2009 to March 2011, ERAS.

Response Figure R5. Trends in percentage of index of multiple deprivation category following primary hip replacement, in England, UK, 2008 – 2016, by month.

Index of multiple deprivation, IMD. Grouped from “IMD Decile Group” variable using the IMD Overall Ranking to identify which one of five groups a Super Output Area belongs to (rank of 32482 is the least deprived, and 1 the most deprived). Least deprived 20%, 25987-32482; Less deprived 20-40%, 19490-25986; Less deprived 40-60%, 12994-19489; More deprived 20-40%, 6497-12993; Most deprived 20%, 1-6496. Enhanced recovery after surgery programme implemented in England from April 2009 to March 2011, ERAS.

Response Figure R6. Trends in percentage of surgical approach category following primary hip replacement, in England, UK, 2008 – 2016, by month.

% of surgical approach by month in hip replacement

Other surgical approach category includes: anterior, antero-lateral, hardinge, lateral, trochanteric osteotomy, and other. Enhanced recovery after surgery programme implemented in England from April 2009 to March 2011, ERAS.

In the discussion section of the paper we had attempted to comment on these “External influencing factors” about decreasing numbers of available beds and operating theatres (page 19/last paragraph/lines 456-462): “ There are fewer hospital beds available for hip replacement today than at the beginning of the study period, and wards have been closed. For example, there was an average daily 10,015 occupied beds open overnight for trauma and orthopaedics in England between April and June 2010, dropping to 8,770 between October and December 2016(43). Conversely, the number of primary hip replacements in England increased from 67,128 in 2008 to 87,733 in 2016(6). It has been estimated that 97,516 total hip replacements will take place in 2035(44).”

Author action: We have added the following text to the discussion (page 20/lines 471-474): “ERAS has kept the improvements happening when other changes were occurring which may have caused deterioration, e.g.: older, sicker, and more obese patients. However, changes in the case-mix of patients have not altered improving trends in outcomes of surgery”

Reviewer: 2

Reviewer Name

THOMAS WAINWRIGHT

Institution and Country

Bournemouth University
UK

Please state any competing interests or state 'None declared':

Thomas Wainwright has received speaker's honoraria for various enhanced recovery after surgery symposia but has no relevant conflict of interest related to this work. He is a director/treasurer of The Enhanced Recovery after Surgery Society (UK) c.i.c. (not-for-profit organization—Company No. 10932208).

Please leave your comments for the authors below

Thank you for inviting me to review this work.

The authors have done an excellent job and I think that this is an important research question which the authors have answered thoroughly.

I have one point that I feel is important to make clear and to be addressed, in both the methods, and in the discussion on outcomes. During the time period studied, there has been an increasing amount of NHS hip replacements performed in non-NHS settings as either waiting list initiative work, or through patient choice and the AQP route. Whilst these pts may still be registered on the NJR they will not to my knowledge appear in HES statistics. Please can the authors be explicit on how this has been handled within the data, and also consider the effects on outcomes that this may of had in the discussion. If this point is addressed I would recommend for publication.

Author response: We thank the reviewer for this helpful comment and agree that this is an important point. Through linkage of data to the English Hospital Episode Statistics (HES), private patients are excluded, as HES data are only collected for patients funded by the NHS, however this does mean that we do capture NHS funded patients treated in private hospitals. We have information about the place where patients were operated in our linked dataset NJR-HES-PROMs. This variable is gathered in the NJR registry. This includes organisations in both the NHS and the independent healthcare sector.

Unit Type	Freq.	Percent
Public hospital	335,921	76.53
Independent hospital	80,938	18.44
Independent sector treatment centre (ISTC)	22,062	5.03
Total	438,921	100

The 5% for hip replacements treated in independent sector treatment centres (ISTC) is mentioned in Cooper et al.⁹ Between 2008 and 2016, the monthly mean percentage of patients operated on in public hospitals decreased, while those in independent hospitals increased. Patients treated in ISTC increased between 2008 and 2010. Between 2011 and 2016 the monthly mean percentage of those patients undergoing primary hip replacement in these centres remained steady (See figure below). ISTC opened in 2 waves. For the first wave 23 ISTC centres opened in 2005 or 2006. For our study we do not have information on outcomes of hip replacement for those years (our period study starts in April 2008). During the second wave 9 new centres opened, most of them between 2007 and 2008. We observe an increasing trend in the monthly percentage of hip replacements in ISTCs until 2010.

% of unit type by month in hip replacement

Author action: We have added the following text to the limitations of the study (page 21, lines 495-501) “Although the NJR registry captures all primary hip replacements including those undertaken in the private sector, linkage to English HES data means that we only have access to information on patients receiving NHS funded operations including public and private hospitals. Therefore, this study does not include private funded operations undertaken by the independent sector. It is estimated 13.7% to 19.7% of all hip replacements were carried out by the independent sector in 2012-2013 and 2016-2017, respectively (Source: Hospital Episode Statistics, NHS Digital.)”

In the discussion section on external influencing factors, we also now include the following (page 20, lines 463-471): “During the period of our study we observed an increasing trend in the proportion of NHS funded primary hip replacements being carried out in independent hospitals (increasing from around 10% in 2008 to 27% in 2016) and a small increase in those within Independent Sector Treatment Centres (ISTC) (from 3.5% in 2008 to 5% in 2016). These changes will have supported an increase in capacity for surgery (although such centres typically treat healthier and less complex patients than nearby public hospitals, with a worsening case-mix of those patients treated in public hospitals⁹. Such changes in the sorting of routine and complex patients between public and private hospital settings over time could also influence observed changes in outcomes of surgery over time.”

1. Burn E, Edwards CJ, Murray DW, et al. Trends and determinants of length of stay and hospital reimbursement following knee and hip replacement: evidence from linked primary care and NHS hospital records from 1997 to 2014. *BMJ open* 2018;8(1):e019146. doi: 10.1136/bmjopen-2017-019146 [published Online First: 2018/01/29]
2. Bardram L, Funch-Jensen P, Jensen P, et al. Recovery after laparoscopic colonic surgery with epidural analgesia, and early oral nutrition and mobilisation. *Lancet* (London, England) 1995;345(8952):763-4. doi: 10.1016/s0140-6736(95)90643-6 [published Online First: 1995/03/25]
3. Ljungqvist O, Hubner M. Enhanced recovery after surgery-ERAS-principles, practice and feasibility in the elderly. *Aging Clin Exp Res* 2018;30(3):249-52. doi: 10.1007/s40520-018-0905-1 [published Online First: 2018/02/18]
4. Enhanced-Recovery-Partnership-Programme. Delivering enhanced recovery – Helping patients to get better sooner after surgery. In: Health Do, ed., 2010.

5. Martin S, Smith P. Explaining variations in inpatient length of stay in the National Health Service. *Journal of Health Economics* 1996;15(3):279-304. doi: [https://doi.org/10.1016/0167-6296\(96\)00003-3](https://doi.org/10.1016/0167-6296(96)00003-3)
6. Gaynor M, Ho K, Town RJ. The Industrial Organization of Health-Care Markets. *Journal of Economic Literature* 2015;53(2):235-84. doi: 10.1257/jel.53.2.235
7. Fenn P, Davies P. Variations in length of stay: A conditional likelihood approach. *Journal of Health Economics* 1990;9(2):223-34. doi: [https://doi.org/10.1016/0167-6296\(90\)90019-Y](https://doi.org/10.1016/0167-6296(90)90019-Y)
8. Department_of_Health. Enhanced recovery Partnership Project Report - March 2011. In: NHS-Institute DN-IN, ed.: Department of health, 2011.
9. Cooper Z, Gibbons S, Skellern M. Does competition from private surgical centres improve public hospitals' performance? Evidence from the English National Health Service. *Journal of Public Economics* 2018;166:63-80. doi: <https://doi.org/10.1016/j.jpubeco.2018.08.002>

VERSION 2 – REVIEW

REVIEWER	Bheeshma Ravi University of Toronto, Canada
REVIEW RETURNED	25-Sep-2019

GENERAL COMMENTS	The authors have put in a significant effort and have appropriately answered my queries.
--

REVIEWER	Thomas W. Wainwright Bournemouth University, UK.
REVIEW RETURNED	08-Oct-2019

GENERAL COMMENTS	Thank you for acknowledging my previous review and adding in clarification regarding operations performed in non NHS hospitals. Congratulations on this thorough analysis, it provides useful data and context for future work to improve peri-operative care and outcomes for THR patients.
--